# Farm-Scale Autonomous Welfare Monitoring in Precision Livestock: A Systematic Review of Robotics and Multimodal AI with an Emphasis on the Lab-to-Farm Deployment Gap

## Abstract

While breakthroughs in autonomous robotics and multimodal artificial intelligence (AI) promise continuous, real-time monitoring for precision livestock farming, their practical on-farm application faces significant limitations, revealing a critical lab-to-farm deployment gap. This deployment gap is rooted in fundamental challenges relevant to the embodied AI community: poor model generalization, sim-to-real fragility, and the absence of standardized validation benchmarks. The primary objective of this systematic review is to highlight state-of-the-art knowledge on these technologies to understand and bridge this gap, proposing a path forward that benefits both agricultural practice and on-farm research. From a pool of over 900 articles reviewed on autonomous navigation and AI-driven analytics, we systematically selected 33 studies to propose recommendations for adopting farm-scale autonomous monitoring in precision livestock. Our review reveals that while foundational technologies are well-established, research remains fragmented and often limited to laboratory simulations or small, single-farm field trials. Based on these findings, we propose a deployment-oriented roadmap with recommendations for developing integrated, robust, and scalable systems. Furthermore, we pinpoint a critical deficiency, that is, the lack of a standardized learning representation (ontology/schema) for collected welfare insights. This deficiency prevents the creation of reproducible datasets for the embodied AI community, hindering the development of truly robust and generalizable models for livestock welfare.

## 1 Introduction

The convergence of advanced technology in agriculture has given rise to **precision livestock farming (PLF)**, a field dedicated to optimizing animal health and productivity. A key application within PLF is automated **welfare monitoring**, where technologies such as **artificial intelligence (AI)** and **autonomous robotics** are used to continuously assess animal well-being. The rapid advancement of these robotic and AI systems is expanding the capabilities of PLF, particularly in proactive dairy cattle welfare monitoring (Kate & Neethirajan, 2025; Ferreira & Dórea, 2025). Over the past decade, sensing technologies like computer vision and accelerometers have proven effective at detecting health issues earlier than manual observation (Feighelstein et al., 2024; Tran et al., 2025; Uddin, 2024). Despite these promising results in controlled settings, a significant bottleneck remains in developing robust, generalizable, and cost-effective systems for practical on-farm deployment (Govinda et al., 2025; Myat Noe et al., 2025). Technologies often fail to scale from single-farm trials to diverse, dynamic agricultural environments, hindered by challenges in navigation, data fusion, and edge computing (Damjanović et al., 2025; Sousa et al., 2025).

This systematic review addresses the critical implementation gap with a consolidation of recent research across three interrelated domains, namely, (1) autonomous navigation for agricultural robotics, (2) AI-driven welfare analytics using sensors such as vision and thermal cameras, and (3) multimodal decision support systems. By reviewing 33 studies published between 2021 and 2025, our investigation is guided by a two-part research question: **1) What are the primary technical and methodological obstacles preventing the farm-scale deployment of autonomous welfare mon-**

**itoring systems? and 2) What roadmap can guide the development of standardized learning representations to build a reproducible welfare knowledge base?**

Our main contributions are:

- A systematic synthesis of the state-of-the-art in autonomous robotics and multimodal AI for dairy cattle welfare, identifying key technological trends and methodologies.

- A critical review of the deployment gap between controlled research and practical farm-scale implementation, highlighting persistent challenges in generalization, robustness, and system integration.

- The proposal of a deployment-oriented roadmap with concrete recommendations for future research, focusing on standardized benchmarks, multi-farm validation, and human-in-the-loop systems to accelerate adoption.

- Highlighting the critical need for a standardized learning representation of welfare insights to foster reproducible datasets and accelerate the development of robust models for the embodied AI community.

## 2 FOUNDATIONS IN ROBOTIC AUTONOMY AND AI-DRIVEN WELFARE MONITORING

### 2.1 ADVANCES IN AUTONOMOUS NAVIGATION AND ROBOTICS

The foundation for autonomous farm-scale monitoring is built upon extensive research in mobile robotics and AI-driven navigation. In a comprehensive survey, Damjanović et al. (2025) taxonomized Simultaneous Localization and Mapping (SLAM), a method for a robot to construct a map of an unknown environment while simultaneously tracking its own location within it, and machine learning approaches for indoor mobile robots, highlighting the increasing trend of ML-augmented SLAM while noting its limitations in dynamic and cluttered spaces. Similarly, Govinda et al. (2025) provided an extensive review of Deep Reinforcement Learning (DRL) applications across autonomous systems, a subfield of machine learning where an agent learns optimal behaviors through trial and error to maximize a cumulative reward. They identified the sim-to-real gap as a primary obstacle to practical deployment, a challenge directly relevant to agricultural settings.

Efforts to bridge this gap include novel approaches like the zero-shot DRL for mapless navigation presented by Sivashangaran (2024), which offers a promising alternative to traditional SLAM by enabling generalization to new environments without retraining. Other researchers have focused on integrating classical control with modern AI; for instance, Munaf & Almusawi (2024) demonstrated a hybrid system combining Q-learning, a model-free reinforcement learning algorithm that learns the value of an action in a particular state, with a PID (Proportional-Integral-Derivative) controller, a common feedback control mechanism, for robust trajectory tracking. The practical implementation of these concepts on low-cost hardware like the TurtleBot3 and Raspberry Pi has been explored by Babu et al. (2024) and Ahmed et al. (2025), respectively, showcasing the feasibility of deploying autonomous navigation in constrained settings.

Further exploring DRL, Mukherjee et al. (2025) compared Q-learning, SARSA (State-Action-Reward-State-Action), another on-policy reinforcement learning algorithm, and DQN (Deep Q-Network) algorithms, which use neural networks to approximate the Q-value function, for path planning, confirming that DQN offers greater efficiency in dynamic scenarios. Valcourt et al. (2024) took this a step further by implementing a Q-learning and object detection (YOLOv9) system on a physical robot, demonstrating strong performance in real-time obstacle avoidance. The challenge of exploration in unknown environments was tackled by Alahdal et al. (2025), who used a Value Iteration algorithm within a ROS framework to enhance a robot's ability to autonomously explore and map new spaces.

The fusion of multiple sensor modalities for robust perception is another critical area. Sousa et al. (2025) presented a methodology for integrating wheel odometry, LiDAR (Light Detection and Ranging), which uses lasers to measure distances, and RGB-D (Red-Green-Blue-Depth) cameras, which provide both color and depth information, on a ground robot, enhancing long-term localization and mapping. Sun et al. (2024) developed a similar multi-sensor framework specifically for quadruped

robots, enabling stable navigation in complex, unstructured terrains. Vision-based navigation has also seen significant progress. Zhang et al. (2024a) proposed a Sim2Real domain adaptation method using CycleGAN to bridge the visual gap between simulation and reality, significantly improving performance. Chen et al. (2024) provided a survey of sensor technologies, suggesting an optimal fusion of vision, using models like YOLOv5 (You Only Look Once, version 5) for real-time object detection, and LiDAR for efficient path planning. For perception-driven control, Acquaah et al. (2025) integrated YOLOv5 with a 3D-depth camera for a rule-based collision avoidance system, while Selvanathan et al. (2024) developed an IoT-enhanced path planning system that uses D* Lite and DRL to adapt to dynamic obstacles.

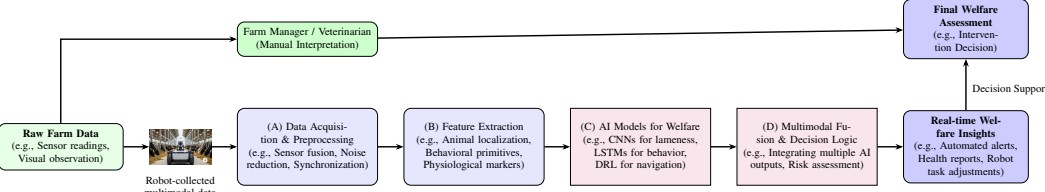

Figure 1: Conceptual diagram of an autonomous dairy cattle welfare monitoring system, integrating multimodal AI with human-in-the-loop verification. The lower path represents the automated system, while the upper path illustrates traditional manual interpretation, with the automated system providing decision support.

## 2.2 AI-Powered Welfare Analytics in Livestock

In parallel with advances in robotics, the field of AI-driven welfare analytics has matured significantly. The application of infrared thermography (IRT), a non-invasive technique that captures thermal radiation to visualize temperature patterns, has been validated by Feighelstein et al. (2024) for the pre-clinical detection of digital dermatitis and further supported by the review from Uddin (2024), which frames IRT as a cornerstone of modern animal health assessment.

For behavioral monitoring, researchers have successfully applied advanced deep learning models to sensor data. El moutaouakil & Falih (2025) and Tran et al. (2025) both showed that Transformer, a deep learning architecture that uses a self-attention mechanism to process sequential data, and LSTM (Long Short-Term Memory) networks, a type of recurrent neural network adept at learning long-term dependencies, respectively, can accurately classify cattle behaviors from accelerometer data, demonstrating the power of sequence modeling for capturing temporal dependencies in animal movement.

The fusion of multiple data sources for a more holistic assessment is a growing trend. In a large-scale study, Dervić et al. (2024) improved lameness detection by integrating sensor data with farm management and weather information across 44 farms. Expanding on this concept, Ferreira & Dórea (2025) and a dissertation sourced from Pro (nodate) conceptualize systems that integrate computer vision, large language models (LLMs), AI systems trained on vast text datasets to understand and generate human-like language, and structured farm data for comprehensive decision-making and phenotype prediction.

Foundational to many of these analytics is robust computer vision. Myat Noe et al. (2025) engineered a YOLOv8-based (You Only Look Once, version 8) multi-camera system for tracking black-coated cattle in challenging open-ranch settings, demonstrating high tracking accuracy on edge devices. For individual identification, Huang et al. (2024) and Cheng et al. (2024a) have developed lightweight models for real-time facial recognition of goats, sheep, and cows. The broader application of machine vision for intelligent robots, including feature extraction and data fusion, was reviewed by Cheng et al. (2024b). Other works, like that of P (2024), have surveyed the use of classic machine learning algorithms such as SVM (Support Vector Machine), a supervised model that finds an optimal boundary between data classes, and Random Forest, an ensemble method that builds multiple decision trees, for general cattle disease prediction from varied data sources. The innovative use of bioacoustics, the scientific study of animal sounds, and video analysis to decode bovine communication and emotional states was proposed by Kate & Neethirajan (2025), opening a new frontier in welfare monitoring. Finally, specialized applications such as the YOLOv5s-based system

(a smaller, faster variant of the YOLOv5 object detector) by Ding et al. (2024) for pipe network inspection showcase how vision technologies developed for industrial settings can be adapted for monitoring farm infrastructure. The convergence of these specialized works in navigation and welfare analytics paves the way for the truly integrated, multimodal systems envisioned by researchers like Zhang et al. (2024b), who are designing platforms that fuse heterogeneous sensor data for robust perception in complex, real-world environments.

# 3 METHODOLOGY OF THE SYSTEMATIC REVIEW

## 3.1 ELIGIBILITY CRITERIA AND SEARCH STRATEGY

The eligibility criteria for this review mandated that included studies be peer-reviewed scholarly articles published between **January 2021 and May 2025**. The substantive focus was required to be on the development or application of autonomous systems and real-time monitoring technologies for dairy cattle welfare. This scope encompassed research on robotics (e.g., SLAM, reinforcement learning), various sensing modalities (e.g., computer vision, thermal imaging, accelerometers), and the use of AI analytics for welfare-related outcomes such as lameness detection or behavior recognition. **Crucially, studies were also included if they focused on other livestock or general indoor robotics (e.g., pipe inspection, multi-robot systems) but presented novel, critical solutions for robustness, edge-compute feasibility, or navigation strategies that are directly transferable to the dynamic farm environment.** Studies were excluded if they did not provide empirical data, were not specific to dairy cattle unless meeting the transferability criteria above, or were unavailable in full-text form.

A comprehensive search was executed across several major academic databases, including Google Scholar, IEEE Xplore, and ScienceDirect. The search query was constructed by combining keywords along three conceptual axes: (1) autonomous robotics, (2) dairy welfare AI, and (3) multimodal systems and edge deployment. To ensure exhaustive coverage, this database search was supplemented with forward and backward citation chasing of key articles. The full, reproducible search string is provided in Appendix A.1.

## 3.2 STUDY SELECTION AND SYNTHESIS

Following the initial search, all identified records were imported into a bibliographic manager for de-duplication. Reviewers then conducted an initial screening of titles and abstracts, which was followed by a full-text assessment of potentially relevant articles against the predefined eligibility criteria. To ensure methodological rigor and mitigate potential bias, a verification protocol was instituted wherein two of the reviewers, the most experienced, assessed a random 20% sample of excluded studies to confirm the consistent application of the criteria. **This verification achieved an inter-rater agreement of** $95\%$**, thereby substantiating the rigorous application of our inclusion/exclusion rules.**

Data from the 33 studies that met the inclusion criteria were subsequently extracted into a structured data registry. Given the significant heterogeneity in study designs, outcome measures, and reported metrics across the selected literature, a quantitative meta-analysis was deemed inappropriate. Consequently, a narrative synthesis was performed. The findings from this synthesis were organized thematically to identify cross-cutting challenges, technological trends, and emergent research opportunities, thereby constructing a coherent overview of the field's current state and future directions.

# 4 RESULTS: A THEMATIC SYNTHESIS

Our analysis of the 33 included studies reveals three primary thematic areas that delineate the current research landscape. The study selection process is visually summarized in a PRISMA diagram (Figure 2), while the distribution of research across different domains and validation settings is illustrated in an evidence map (Figure 3) and a detailed taxonomy (Figure 4).

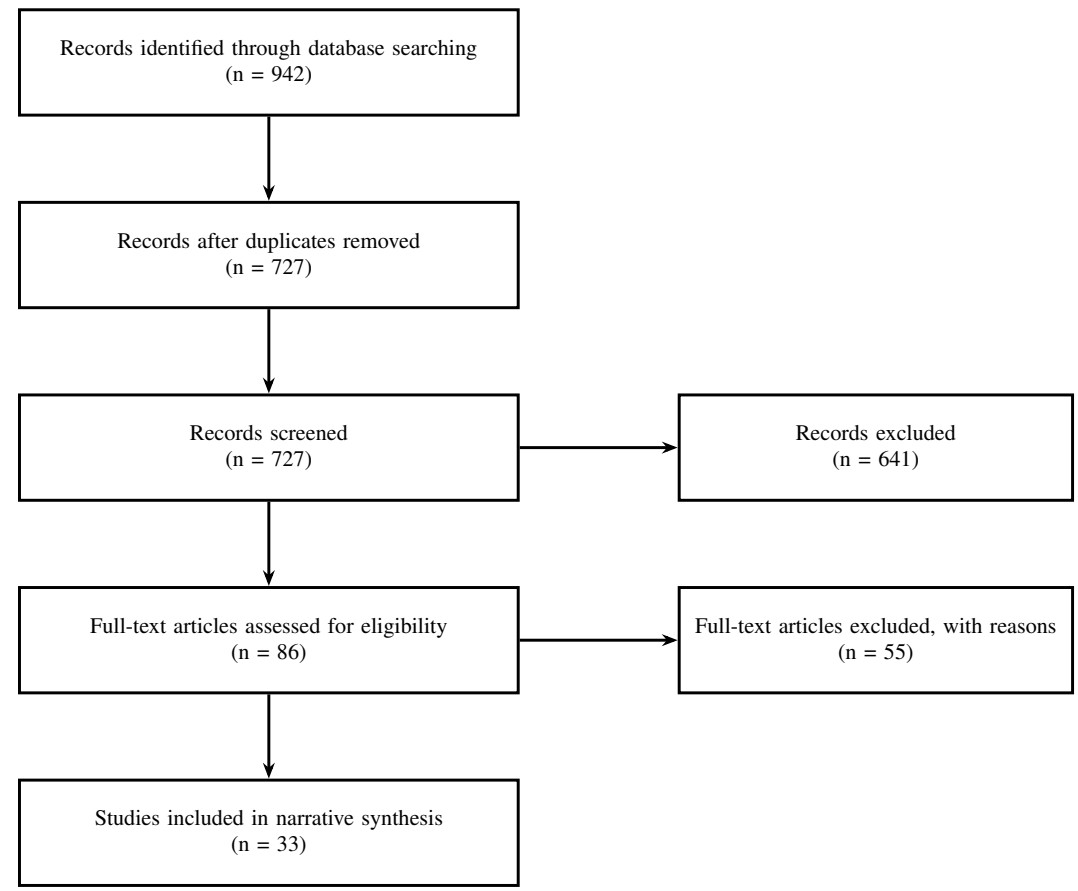

Figure 2: PRISMA flow diagram summarizing the study identification, screening, and inclusion process.

### 4.1 QUANTITATIVE TRENDS IN VALIDATION SETTINGS

To empirically ground the narrative synthesis, we extracted and analyzed the validation settings for the 33 included studies. Our analysis strongly supports the existence of the lab-to-farm deployment gap:

- **Limited Field Validation:** Only 14 of the 33 studies (42%) reported validation in real-world farm ('Field') environments (see Table 1).
- **Simulation/Lab Focus in Robotics:** Of the 12 studies focused purely on Robotics & Autonomy, 8 (67%) were validated exclusively in 'Simulation' or constrained 'Lab' settings, confirming the severity of the sim-to-real challenge.
- **Multimodal Fragmentation:** Studies utilizing Multimodal Systems (N = 4) were highly fragmented across validation settings (Sim = 2, Field = 2), highlighting the nascent state of integrated, robust systems.

These figures substantiate our claim that while foundational technology exists, the majority of research remains in controlled, non-generalizable environments.

### 4.2 THEME 1: AUTONOMY FOR FARM ENVIRONMENTS.

A substantial body of the reviewed literature is dedicated to enhancing robotic autonomy within complex agricultural settings. Researchers frequently employ deep reinforcement learning (DRL) for mapless navigation, enabling robots to operate without pre-existing spatial data (e.g., (Kin et al., 2025), (Sivashangaran, 2024)). Concurrently, multi-sensor Simultaneous Localization and Mapping

(SLAM) is being leveraged to improve mobility and robustness in cluttered farm environments (e.g., (Damjanović et al., 2025), (Opt, 2024)). Although these methodologies demonstrate considerable success in simulated contexts, a significant challenge persists in the sim-to-real transfer, as systems often exhibit fragility when confronted with the dynamic and unpredictable conditions of a working farm, such as the movement of animals or variable lighting (Zhang et al., 2024a).

### 4.3 THEME 2: AI FOR WELFARE ANALYTICS.

This theme encompasses the application of artificial intelligence to interpret sensor data for animal health monitoring. Infrared thermography (IRT), for instance, has been effectively utilized for the pre-clinical detection of pathologies such as digital dermatitis (Feighelstein et al., 2024). Furthermore, models incorporating data from accelerometers and computer vision have achieved high accuracy in classifying key cattle behaviors that are indicative of welfare states, including those related to lameness and mastitis (Tran et al., 2025), (Ding et al., 2024). A primary limitation within this domain, however, is the issue of external validity. The majority of these studies are conducted on single farms with limited sample sizes, which severely constrains the generalizability of their findings to different cattle breeds, seasonal conditions, or barn architectures (Dervić et al., 2024).

### 4.4 THEME 3: MULTIMODAL FUSION AND DECISION SUPPORT.

The integration of heterogeneous data streams—including imagery, accelerometry, audio, and structured farm records—has been shown to improve the reliability and timeliness of welfare monitoring systems (Ferreira & Dórea, 2025), (Pro, nodate). Despite its potential, progress in this area is constrained by the absence of standardized metrics and annotation protocols. Moreover, while the objective is often on-robot deployment, few studies systematically evaluate or report the performance of their fusion models in terms of latency and resource consumption on the embedded, edge-compute platforms that are characteristic of farm-scale robotics.

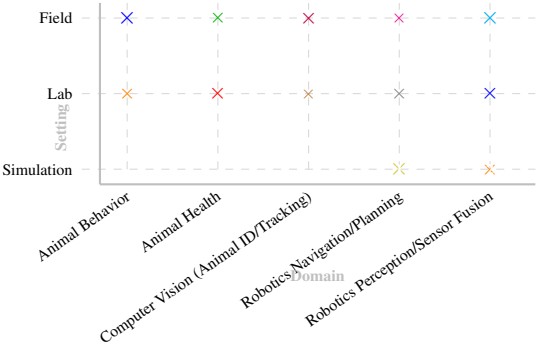

Figure 3: Evidence map by domain × setting.

## 5 DISCUSSION: THE PATH TO FARM-SCALE DEPLOYMENT

Our synthesis of the literature reveals that while the constituent components for autonomous welfare monitoring are largely well-developed, the primary impediment to widespread adoption is the challenge of scaling these technologies from controlled trials to robust, farm-wide implementations. The current research landscape is characterized by a prevalence of single-site studies, a lack of consistent evaluation metrics, and limited transparency regarding the costs and reliability of these systems. These factors collectively hinder their broader adoption. In the following subsections, we distill the key insights from our findings and propose a deployment-oriented roadmap to address these critical challenges.

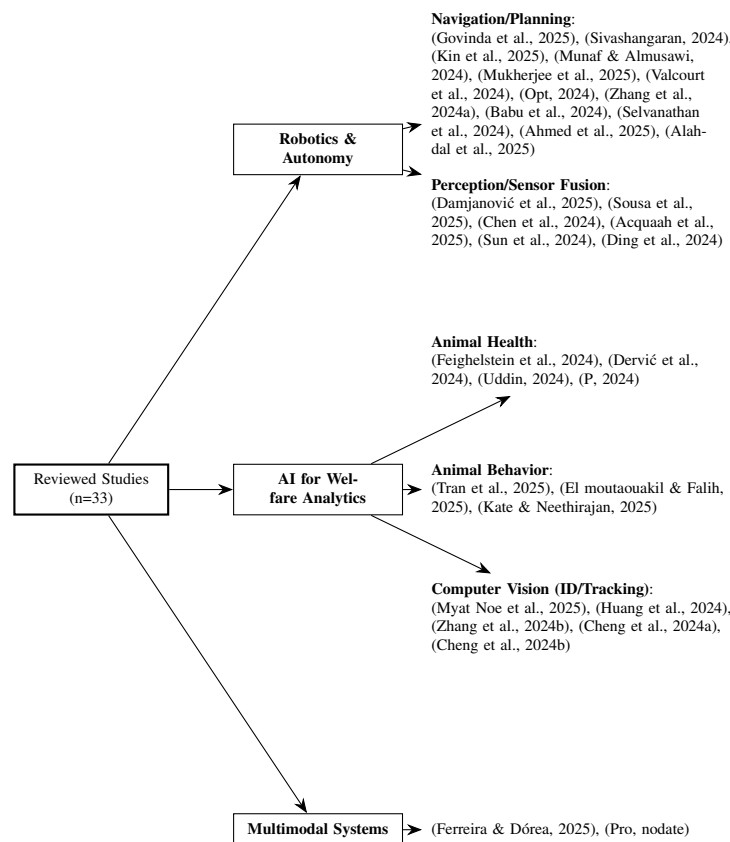

Figure 4: A taxonomy of the 33 reviewed studies, organized by primary research domain.

## 5.1 KEY INSIGHTS FROM THE REVIEW

The evidence base demonstrates a significant disconnect between research in robotics autonomy, which is frequently confined to simulations, and research in welfare analytics, which, while often conducted on-farm, is typically limited in scale. Navigation systems that exhibit high efficacy in controlled settings often fail when confronted with the dynamic and unpredictable realities of a working farm. Similarly, welfare analytics models trained on data from a single herd rarely generalize to other farms without extensive retraining and recalibration. **A recurring theme throughout the literature is the insufficient focus on system-level integration. This fragmentation of research efforts (as empirically shown in Section 4.1) represents the single greatest barrier to practical, large-scale deployment, as few studies present a truly end-to-end system that combines autonomous mobility, multimodal perception, and real-time analytics on edge-constrained hardware.**

## 5.2 A ROADMAP FOR FUTURE RESEARCH

To bridge the deployment gap identified in this review, we propose a deployment-oriented research roadmap founded on three key pillars:

- **Establishment of Standardized Benchmarks and Multi-Farm Datasets:** The research community urgently requires standardized, open-access benchmarks for both robotics and welfare analytics within agricultural contexts. This necessitates the creation of large-scale, multi-farm, and multi-season validation datasets that capture the full spectrum of operational diversity (e.g., different breeds, barn architectures, and climatic conditions). Such resources are indispensable for the development and rigorous validation of models that are truly generalizable. **Validation must move beyond simple accuracy (Acc) and explicitly include deployment-critical factors such as cross-farm generalization loss ($\mathcal{L}_{\mathbf{gen}}$) and**

**edge-compute performance:**

$$\mathcal{M} = (\mathcal{L}_{\text{gen}}, \text{Latency}_{\text{edge}}, \text{PowerConsumption}).$$

- **A Shift Toward Edge-Aware and Integrated Systems:** Future research must pivot from developing isolated "point models" to engineering integrated, end-to-end systems. This paradigm shift requires an "edge-aware" design philosophy, wherein models are developed with explicit budgets for computation, power, and latency (e.g., ¡100ms inference, ¡ 5W consumption **?**). A focus on techniques such as model compression, data-efficient training, and federated learning is essential to ensure practical on-device performance.

- **Implementation of Human-in-the-Loop Verification:** To foster trust and facilitate farmer adoption, initial deployments should incorporate human-in-the-loop verification frameworks. These systems should not be envisioned as fully autonomous decision-makers but rather as powerful assistive tools that flag animals requiring attention and provide supporting evidentiary data (e.g., video clips, thermal images) to a veterinarian or farm manager. This approach not only provides a crucial fail-safe but also establishes a valuable feedback loop, where expert-validated data can be used to continuously retrain and refine the underlying AI models.

- **Standardizing Welfare Data as a Learnable Representation:** The review highlights a fundamental obstacle: the absence of a standardized learning representation for collected welfare insights. **This representation must take the form of an underlying ontology or schema (see Appendix B) that links heterogeneous multimodal inputs to a common, machine-readable welfare output.** This gap prevents the creation of a useful knowledge base for the embodied AI community and hinders the development of robust models. To move forward, research must focus on translating these insights into standardized, learnable formats. Establishing this common representation is a crucial step toward creating the reproducible datasets required to train the next generation of robust models for livestock welfare **??**. **Furthermore, we advocate for the use of Generative AI (e.g., GANs, Procedural Content Generation) to realize this goal by synthetically normalizing and augmenting the limited, heterogeneous on-farm datasets, leveraging established techniques from the broader robotics community (e.g., Sim2Real domain adaptation ?).**

## 5.3 LIMITATIONS OF THIS REVIEW

This review's methodology is subject to several limitations centered on our selection protocol. First, the protocol was not pre-registered, which may reduce the transparency of the review process.

Furthermore, the protocol for study selection and data extraction was executed by the reviewer-researchers, which introduces a potential for subjective bias. Although the structured protocol and explicit inclusion criteria were designed to ensure consistency and mitigate this risk, the human element in this process means that the application of these criteria may not have been perfectly uniform, potentially affecting the final set of included studies.

Finally, a key limitation emerged as a direct consequence of our selection protocol: the included studies exhibited significant heterogeneity in their methodologies and reported metrics. This diversity made a quantitative meta-analysis inappropriate, thus necessitating the narrative synthesis approach used in this paper, which can be more influenced by the authors' interpretations.

## 6 POSITIONING WITHIN EXISTING LITERATURE

While several reviews have addressed the application of AI in precision livestock farming (PLF) (Lee & Kim, 2024), they have typically focused on a single modality, such as computer vision for lameness detection (Smith & Jones, 2022). **Older, broader surveys, such as the general robotics review by Patel & Chen (2023) and related pre-2021 works, typically did not capture the impact of modern DRL or Transformer architectures on deployment feasibility.** The present work is distinct in its specific synthesis of the integration challenges and the deployment gap for systems that are concurrently *autonomous, mobile*, and utilize *multimodal* data for the purpose of real-time welfare monitoring. **The strict 2021-2025 time window was chosen to focus on the essential 'AI-convergence' period where breakthroughs in Deep Learning (e.g., YOLOv8, modern DRL)**

**and low-cost embedded hardware began to make farm-scale autonomous deployment practically feasible.** By analyzing the critical intersection of robotics, AI-driven analytics, and the practicalities of on-farm deployment, this review provides a unique, systems-level perspective aimed at bridging the gap between academic research and practical implementation.

# 7 CONCLUSION

This systematic review has established that the core technologies for autonomous dairy cattle welfare monitoring are largely in place. The synthesis of the literature indicates that foundational methods in computer vision, sensor analytics, and mobile navigation are sufficiently mature to effectively detect welfare issues, often with greater timeliness than traditional manual observation. Modern navigation techniques, in particular, are increasingly enabling mobile systems to operate safely and effectively within the dynamic conditions of real-world farm environments.

Despite this technological readiness, a significant challenge remains: technologies validated in controlled settings frequently yield inconsistent or unreliable results in operational farm environments. The primary obstacle is therefore not one of invention but of implementation—specifically, scaling from controlled experimental trials to robust, farm-wide systems. The current body of research is dominated by single-site studies, inconsistent evaluation metrics, and a lack of transparency regarding system costs and long-term reliability, all of which hinder the broader adoption of these technologies.

This review puts forth a deployment-oriented roadmap designed to address these challenges. We contend that future success will depend on a collective effort to establish standardized validation datasets, a dedicated research focus on creating efficient and integrated end-to-end systems, and the implementation of human-centric designs that build trust with end-users. By following this roadmap, the research community can accelerate the ethical, profitable, and scalable adoption of these transformative technologies, which stand to significantly improve both animal welfare and agricultural productivity.

Distinct from prior work, which has typically reviewed either single AI modalities in PLF or agricultural robotics more broadly, the present review provides a unique contribution. Its originality stems from a tripartite focus: first, a specific synthesis of the challenges defining the lab-to-farm deployment gap; second, the formulation of a deployment-oriented roadmap with concrete recommendations to bridge this gap; and third, its identification of a critical and foundational need for a standardized learning representation of welfare insights to enable a reproducible knowledge base for the embodied AI community.

## ETHICAL CONSIDERATIONS

The deployment of the technologies reviewed in this paper carries significant ethical responsibilities. Key considerations include the privacy and security of sensitive farm and animal data. Furthermore, the risk of algorithmic bias is substantial; models trained on data from specific breeds or farm types may perform inequitably on others, potentially leading to disparities in health outcomes. The often opaque nature of "black-box" AI models presents a challenge to accountability and trust, underscoring the necessity for explainable AI (XAI) in systems intended for use by veterinarians and farm managers. Additionally, the socio-economic impact on farm labor must be carefully considered. We advocate for a focus on systems that augment, rather than replace, human expertise, ensuring that technology serves as a tool to enhance animal welfare, not merely to optimize production.

## REPRODUCIBILITY STATEMENT

We are committed to ensuring the reproducibility of this systematic review. Our complete methodology, including the databases searched, the eligibility criteria for inclusion and exclusion, and the multiple-reviewer screening protocol with its verification checks, is described in detail in Section 3. To allow for the direct replication of our literature search, the full search string utilized for the IEEE Xplore database is provided in Appendix A.1. Furthermore, the complete data extraction spreadsheet, which contains the data logged for all 33 included studies, is available as part of the supplementary materials. Together, these resources provide a transparent and verifiable account of our review process, from the initial search to the final synthesis.

AUTHOR CONTRIBUTIONS AND DISCLOSURES

The authors utilized a large language model to aid in refining the language and structure of the title, abstract, and various sections, as well as to assist in formatting the manuscript according to the conference template and BibTeX standards.

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

# A APPENDIX

## A.1 SEARCH STRING

The following search string was adapted for each database. The version below is for the IEEE Xplore digital library, executed on June 5, 2025. The query combines keywords along the three primary conceptual axes of the review using Boolean operators.

```
(
  % Axis 1: Autonomous Robotics
  ("autonomous robot" OR "mobile robot" OR "unmanned ground vehicle"
  OR "UGV" OR "SLAM" OR "simultaneous localization and mapping"
  OR "reinforcement learning" OR "mapless navigation")
  AND
  % Axis 2: Dairy Welfare & AI
  ("dairy cattle" OR "cow" OR "bovine" OR "livestock")
  AND
  ("animal welfare" OR "lameness detection" OR "mastitis"
  OR "behavior monitoring" OR "health monitoring"
  OR "precision livestock farming")
  AND
  % Axis 3: Multimodal & Edge Deployment
  ("computer vision" OR "thermal" OR "infrared thermography"
  OR "accelerometer" OR "sensor fusion" OR "multimodal AI"
  OR "edge computing" OR "edge AI")
)
```

## A.2 Detailed Study Characteristics

Table 1: Literature review (Part 1: Study Setup)

| Citation | Year | Journal/ Conf. | Math | Variables | Temp. | Sim | Lab | Field | Dataset sources |
|---|---|---|---|---|---|---|---|---|---|
| Damjanović et al. Damjanović et al. (2025) | 2025 | Mach. Vis. Appl. | MSE, RMSE, MAE | RGB, depth, LiDAR, IMU → mapping | No | ✓ | ✓ | ✓ | Public datasets |
| Govinda et al. Govinda et al. (2025) | 2025 | IEEE Trans. ITS | DRL rewards | images, Li-DAR, GPS → navigation | No | ✓ | ✓ | ✓ | Survey of papers |
| Feighelstein et al. Feighelstein et al. (2024) | 2024 | Sci. Rep. | accuracy, precision | thermal hoof images → DD prediction | Yes | ✗ | ✓ | ✓ | Proprietary data |
| El Moutaouakil & Falih El moutaouakil & Falih (2025) | 2025 | Innov. Tech. | transformer | accelerometer → behavior | Yes | ✗ | ✗ | ✓ | Collar sensor data |
| Huang et al. Huang et al. (2024) | 2024 | Int. J. Patt. Rec. AI | YOLOv5+FPN | bounding-box → face detection | | ✗ | ✗ | ✗ | ✓ | Goat/sheep images |
| Sivashangaran Sivashangaran (2024) | 2024 | VT thesis | DDPG, TD3, SAC | LiDAR, RGB-D → actions | Yes | ✓ | ✓ | ✓ | In-house sim. logs |
| ProQuest (n.d.) Pro (nodate) | n.d. | ProQuest thesis | reviewed metrics | images/video/accel → health | ✗ | ✗ | ✗ | ✓ | Literature datasets |
| Kate & Neethirajan Kate & Neethirajan (2025) | 2025 | bioRxiv | ANOVA, t-SNE | acoustic+video → ingestive | Yes | ✗ | ✓ | ✓ | Public audio data |
| Babu et al. Babu et al. (2024) | 2024 | IEEE SPICES | ROS2 Nav2 | LiDAR, odom. → path | ✗ | ✓ | ✗ | ✓ | Gazebo + Robot logs |
| Mukherjee et al. Mukherjee et al. (2025) | 2025 | IEEE IEMENTech | DDPG, PPO, SAC | LiDAR/camera → actions/maps | ✗ | ✓ | ✗ | ✓ | Sim + Robot logs |
| Dervić et al. Dervić et al. (2024) | 2024 | J. Dairy Sci. | RF, Boruta, PDPs | multi-source data → lame-ness | ✗ | ✗ | ✓ | ✓ | D4Dairy: 44 farms |
| Uddin Uddin (2024) | 2024 | Animal Sci. Cases | regression | IRT, cortisol, etc. → stress | Yes | ✗ | ✗ | ✓ | IRT + physio. data |
| Acquaah et al. Acquaah et al. (2025) | 2025 | SoutheastCon | YOLOv5 CIoU | image+depth → detection | ✗ | ✓ | ✗ | ✓ | Indoor nav. logs |
| Sousa et al. Sousa et al. (2025) | 2025 | IEEE ICARSC | 2D/3D SLAM | odom., laser, LiDAR → pose | ✗ | ✓ | ✗ | ✓ | logs + open-source |

*continued from previous page*

| Citation | Year | Journal/ Conf. | Math | Variables | Temp. | Sim | Lab | Field | Dataset sources |
|---|---|---|---|---|---|---|---|---|---|
| Munaf & Almusawi Munaf & Almusawi (2024) | 2024 | JESA | Q-learning, PID | state, LiDAR → velocity | ✗ | ✓ | ✗ | ✗ | Simulation only |
| Selvanathan et al. Selvanathan et al. (2024) | 2024 | IEEE ICONAT | DWA + IoT logic | LiDAR, ultrasonic → avoidance | ✗ | ✓ | ✗ | ✓ | Sim + IoT logs |
| Ferreira & Dórea Ferreira & Dórea (2025) | 2025 | J. Dairy Sci. | CNN, RNN, ViT | multimodal → phenotype | ✗ | ✗ | ✓ | ✓ | published datasets |
| Kin et al. Kin et al. (2025) | 2025 | IEEE ROBOTHIA | DQN, PPO | camera, LiDAR → actions | ✗ | ✓ | ✗ | ✓ | Sim + field logs |
| Tran et al. Tran et al. (2025) | 2025 | JMST | LSTM, BPTT | accelerometer → behavior | Yes | ✗ | ✗ | ✓ | real accel. data |
| Sun et al. Sun et al. (2024) | 2024 | IEEE CCC | EKF, graph SLAM | IMU, encoders → mapping | ✗ | ✗ | ✓ | ✓ | quadruped trials |
| Zhang et al. (2024a) Zhang et al. (2024b) | 2024 | IEEE TCDS | transformer, GMM | RGB-D, QR, enc. → pose | ✗ | ✗ | ✓ | ✓ | 50k prop. images |
| Optimal navigation Opt (2024) | 2024 | JST-HaUI | Bellman updates | state, map → Q-values | ✗ | ✓ | ✗ | ✗ | sim. data only |
| P, S. P (2024) | 2024 | IRJET | NB, KNN, SVM | multimodal → disease | ✗ | ✗ | ✓ | ✓ | review datasets |
| Ding et al. Ding et al. (2024) | 2024 | Springer | YOLOv5s CIoU | boxes, depth → detection | ✗ | ✗ | ✓ | ✓ | pipe inspect. data |
| Cheng et al. (2024a) Cheng et al. (2024a) | 2024 | IEEE CEEML | - | - | ✗ | ✗ | ✗ | ✓ | - |
| Cheng et al. (2024b) Cheng et al. (2024b) | 2024 | IEEE AIDLNN | - | - | ✗ | ✗ | ✗ | ✓ | - |
| Zhang et al. (2024b) Zhang et al. (2024a) | 2024 | IEEE ISPA | CycleGAN | vision → sim2real | ✗ | ✓ | ✗ | ✓ | Sim+real data |
| Chen et al. Chen et al. (2024) | 2024 | App. Comp. Eng. | YOLOv5, RRT | vision, LiDAR → path | ✗ | ✓ | ✗ | ✓ | - |
| Valcourt et al. Valcourt et al. (2024) | 2024 | IEEE ARIS | Q-learning, YOLOv9 | vision, sensors → path | ✗ | ✓ | ✗ | ✓ | Real robot exp. |

| Citation | Year | Journal/ Conf. | Math | Variables | Temp. | Sim | Lab | Field | Dataset sources |
|---|---|---|---|---|---|---|---|---|---|
| Ahmed et al. Ahmed et al. (2025) | 2025 | IEEE ICMLAS | A* algorithm | grid map → path | ✗ | ✗ | ✓ | ✓ | Rover exp. |
| Alahdal et al. Alahdal et al. (2025) | 2025 | IEEE ROBOTHIA | Value Iteration | LiDAR → exploration | ✗ | ✓ | ✗ | ✓ | Real robot exp. |
| Myat Noe et al. Myat Noe et al. (2025) | 2025 | Sci. Rep. | YOLOv7 | 1500 images (prop.) | ✓ | ✗ | ✗ | ✓ | Lightweight models for edge. |

Table 2: Literature review (Part 2: Technical Details & Opportunities)

| Citation | Phys. | Physio. | Chosen data | Hardware | Software | Comp. | Sim | Lab | Field | Research opportunities |
|---|---|---|---|---|---|---|---|---|---|---|
| Damjanović et al. | ✗ | ✗ | Public datasets | Varies by review | ROS & SLAM libs | ✓ | ✓ | ✓ | ✓ | Long-term adaptation. |
| Govinda et al. | ✗ | ✗ | Reviewed papers | Varies; GPU-heavy | PyTorch, TF, ROS | ✓ | ✓ | ✓ | ✓ | Sample-efficient DRL. |
| Feighelstein et al. | ✓ | ✓ | 569 IRT images | IR thermal camera | Python (sci-kit) | ✗ | ✓ | ✓ | ✓ | Expand data; real-time models. |
| El Moutaouakil & Falih | ✗ | ✓ | Accelerometer data | Sensor collars | Python (PyTorch/TF) | ✗ | ✗ | ✓ | ✓ | Multimodal sensing. |
| Huang et al. | ✓ | ✗ | Annotated images | Standard camera | Python, YOLOv5 | ✗ | ✗ | ✗ | ✓ | Real-time embedded detection. |
| Sivashangaran | ✗ | ✗ | LiDAR, RGB-D logs | XTENTH-CAR robot | Python (PyTorch/TF) | ✓ | ✗ | ✓ | ✓ | OOD generalization. |
| ProQuest (n.d.) | ✗ | ✗ | image/video/accel | — | Reviewed lit. tools | ✗ | ✗ | ✗ | ✓ | Integrate unified platforms. |
| Kate & Neethirajan | ✗ | ✗ | Audio datasets | Microphones | Python (sci-kit, LLM) | ✗ | ✓ | ✓ | ✓ | Edge audio decoding. |
| Babu et al. | ✗ | ✗ | Gazebo + TurtleBot3 | TurtleBot3 Burger | ROS2 Nav2 stack | ✓ | ✗ | ✓ | ✓ | Vision-based obstacle detection. |
| Mukherjee et al. | ✗ | ✗ | Sim + real-robot logs | Custom mobile platform | Python (DRL) & SLAM | ✓ | ✗ | ✓ | ✓ | Hybrid SLAM+DRL. |
| Dervić et al. | ✗ | ✓ | Multi-source farm data | AMS equipment | Python, RF, Boruta | ✗ | ✓ | ✓ | ✓ | Real-time lameness alerts. |
| Uddin | ✗ | ✓ | Thermal images + physio | IRT camera, sensors | SPSS, R | ✗ | ✗ | ✗ | ✓ | Extend IRT to other species. |

*continued from previous page*

| Citation | Phys. | Physio. | Chosen data | Hardware | Software | Comp. | Sim | Lab | Field | Research opportunities |
|---|---|---|---|---|---|---|---|---|---|---|
| Acquaah et al. | ✗ | ✗ | Indoor nav. logs | Open-source | YOLOv5, rule-based | ✗ | ✓ | ✗ | ✓ | Unsupervised sim2real. |
| Sousa et al. | ✗ | ✗ | Onboard logs + repo | Hangfa Discovery Q2 | ROS, Gmapping | ✗ | ✓ | ✗ | ✓ | Edge deployment frameworks. |
| Munaf & Almusawi | ✗ | ✗ | Simulation data | Sim. diff. drive robot | MATLAB/Simulink | ✗ | ✓ | ✗ | ✓ | Adaptive PID+DRL. |
| Selvanathan et al. | ✗ | ✗ | ROS+IoT logs | Robot w/ LiDAR & IoT | DWA & IoT logic | ✗ | ✓ | ✗ | ✓ | Cloud multi-robot coordination. |
| Ferreira & Dórea | ✗ | ✓ | Multimodal datasets | – | CNNs, RNNs, ViT | ✗ | ✗ | ✗ | ✓ | Retrieval-augmented selection. |
| Kin et al. | ✗ | ✗ | Sim + field logs | Agri. mobile robot/UAV | DQN, PPO, Bellman | ✗ | ✓ | ✗ | ✓ | Domain-adaptive DRL. |
| Tran et al. | ✗ | ✓ | Accelerometer data | Collar-worn IMUs | LSTM, BPTT | ✗ | ✗ | ✗ | ✓ | On-device sequence analysis. |
| Sun et al. | ✗ | ✗ | Quadruped trials | Quadruped robot | EKF, graph SLAM | ✗ | ✗ | ✗ | ✓ | Dynamic obstacle fusion. |
| Zhang et al. (2024a) | ✗ | ✗ | 50k RGB-D frames | IMR platform + RealSense | Transformers, GMM | ✗ | ✗ | ✗ | ✓ | Long-term consistency. |
| Optimal navigation | ✗ | ✗ | ROS-Gazebo data | Sim-only diff. drive | ROS, RL scripts | ✓ | ✗ | ✗ | ✗ | Formal verification of RL. |
| P, S. | ✗ | ✗ | Review datasets | – | NB, KNN, SVM | ✗ | ✗ | ✗ | ✓ | Cross-species disease models. |
| Ding et al. | ✗ | ✗ | Pipe inspection data | Inspection robot | YOLOv5s, fusion | ✗ | ✗ | ✗ | ✓ | On-pipe autonomous inspection. |
| Cheng et al. (2024a) | ✓ | ✗ | 200 hours video (prop.) | RGB camera | DL (CNNs) | ✗ | ✗ | ✗ | ✓ | Automated ID linking. |
| Cheng et al. (2024b) | ✓ | ✗ | 150 hours video (prop.) | RGB camera | DL (CNNs) | ✗ | ✗ | ✗ | ✓ | Multi-view fusion. |
| Zhang et al. (2024b) | ✗ | ✗ | Sim+real data | Nvidia Jetson (sim2real) | CycleGAN | ✓ | ✓ | ✗ | ✓ | Adversarial domain adaptation. |
| Chen et al. | ✗ | ✗ | Sim+real data | Turtlebot3 | YOLOv5, RRT | ✓ | ✓ | ✗ | ✓ | Robustness to dynamic env. |
| Valcourt et al. | ✓ | ✗ | Real robot exp. | RoboMaster EP Core | Q-learning, YOLOv9 | ✓ | ✓ | ✗ | ✓ | Multi-robot coordination. |
| Ahmed et al. | ✓ | ✗ | Rover exp. | Raspberry Pi Rover | A* algorithm | ✓ | ✗ | ✓ | ✓ | Dynamic re-planning. |

*continued from previous page*

| Citation | Phys. | Physio. | Chosen data | Hardware | Software | Comp. | Sim | Lab | Field | Research opportunities |
|---|---|---|---|---|---|---|---|---|---|---|
| Alahdal et al. | ✓ | ✗ | Real robot exp. | ROS-based robot | Value Iteration | ✓ | ✓ | ✗ | ✓ | Human-robot collaboration. |
| Myat Noe et al. Myat Noe et al. (2025) | ✓ | ✗ | 1500 images (prop.) | Raspberry Pi camera | YOLOv7 | ✗ | ✗ | ✗ | ✓ | Lightweight models for edge. |

# B ILLUSTRATIVE EXAMPLE OF STANDARDIZED WELFARE REPRESENTATION

## B.1 FIRST-CUT SCHEMA FOR STANDARDIZED WELFARE INSIGHT REPRESENTATION

To address the critical need for a standardized learning representation, we provide a first-cut schema (ontology) for connecting heterogeneous sensor data to machine-readable welfare insights. This schema is designed to enable reproducible dataset generation across multiple farms, moving the concept of the Standardized Learning Representation from a vision to a concrete technical proposal.

Table 3: Illustrative Schema for Standardized Welfare Insight Representation

| Field | Data Type | Description / Example Standard |
|---|---|---|
| **Welfare Event ID** | Unique Identifier | Lameness-F1-001, Mastitis-F3-015 |
| **Outcome Label** | Standardized String | LAMENESS (Slight), HEALTHY (Baseline) |
| **Temporal Signature** | Timestamp + Duration | 2025-10-25T14:30:00Z, Duration: 35s |
| **Multimodal Links (Pointers)** | URI/File Path | |
| Visual Ptr | URL / Path | /data/F1/video/cowID_lameness_clip.mp4 |
| Accelerometer Ptr | URL / Path | /data/F1/sensor/cowID_accel_raw.csv |
| **Deployment Context** | String/Enum | |
| Farm Architecture | Enum | TIE-STALL, FREE-STALL |
| Breed/Herd | String | Holstein/Jersey |
| **Validation Status** | Boolean/Enum | VET-CONFIRMED, FARMER-FLAGGED |

This simple representation ensures that disparate data (e.g., raw acceleration signals and high-resolution video) are consistently linked to a common, machine-readable welfare label (Outcome Label) within a standardized deployment context. This directly enables cross-farm generalization training for embodied AI models by standardizing both the output and the contextual metadata.

