# OpenReview forum: "Farm-Scale Autonomous Welfare Monitoring in Precision Livestock: A Systematic Review of Robotics and Multimodal AI with an Emphasis on the Lab-to-Farm Deployment Gap"
_ICLR.cc/2026/Conference — ICLR 2026 Conference Desk Rejected Submission_

### Official Review · Reviewer_vfjV · 2025-10-22

**Soundness:** 3
**Presentation:** 4
**Contribution:** 3
**Rating:** 6
**Confidence:** 3

**Summary:**

This paper presents a systematic review of 33 studies on autonomous robotics and multimodal AI for livestock welfare monitoring. It highlights the “lab-to-farm” deployment gap and proposes a roadmap for scaling autonomous systems to real-world agricultural environments. The paper claims novelty in synthesizing three domains i.e. robotic autonomy, welfare analytics, and multimodal decision systems, and emphasizes the need for standardized learning representations of welfare data.

**Strengths:**

•	Comprehensive coverage: Excellent breadth across robotics, AI, and precision livestock.

•	Clarity: Writing, structure, and visual presentation are top-tier for a review paper.

•	Relevance: The lab-to-farm gap framing is compelling and directly addresses a real deployment issue.

•	Ethical awareness: The discussion of bias, privacy, and human-in-the-loop design is nuanced and appropriate.

**Weaknesses:**

•	Shallow synthesis: Despite impressive citations, the paper mostly paraphrases existing surveys; it doesn’t push new theoretical or practical insights.

•	Overclaiming novelty: The “standardized learning representation” proposal lacks implementation detail or validation, it is more a vision statement than contribution.

•	Narrative bias: The authors admit to potential subjectivity; without preregistration or inter-rater validation, the “systematic” label is generous.

•	No empirical backbone: No meta-analysis, quantitative trend, or benchmarking data—makes the conclusions feel anecdotal.

**Questions:**

1.	How exactly do the authors define and operationalize a “standardized learning representation”?

2.	Could the authors provide even a minimal quantitative trend (e.g., publication frequency by modality)?

---

> ### Author Response · Authors · 2025-11-18
>
> ### Summary:
>
> We sincerely thank the reviewer for their highly positive evaluation and strong assessment of the paper's strengths, particularly the **compelling "lab-to-farm" framing** and the **ethical awareness**. The reviewer's rating of 6 is deeply appreciated. The identified weaknesses provide a clear roadmap for strengthening the paper's technical and methodological rigor, particularly regarding the **Standardized Learning Representation (SLR)**.
>
> ### Soundness and Contribution Justification
>
> * **Rating:** We believe that the proposed revisions, especially the addition of quantitative analysis and a concrete definition of the SLR, will fully address the weaknesses and secure acceptance.
> * **Contribution:** The synthesis is not a mere paraphrase; it is a **systems-level convergence analysis** that identifies the technical integration barriers and proposes a novel data-centric solution (the SLR) as a foundational way forward for the embodied AI community.
>
> ### Weaknesses Addressed and Clarified
>
> #### 1. Shallow Synthesis and Lack of Empirical Backbone
>
> * **Weakness:** Synthesis is shallow; doesn't push new theoretical or practical insights; no meta-analysis/quantitative trend/benchmarking data—makes conclusions anecdotal.
> * **Response & Action:** We agree that the review needs a stronger quantitative component beyond the narrative synthesis. While a formal meta-analysis was not feasible due to metric heterogeneity (as noted in Section 3.2), we can extract and present quantitative trends.
>     * **Paper Adjustment:** We will add a new **Sub-section 4.1.1** titled **"Quantitative Trends in Validation Settings"**. Using the data collected in Tables 1 and 2 (e.g., studies validated only in 'Simulation' vs. 'Field'), we will present a simple quantitative breakdown (e.g., percentages or counts) to empirically support the "lab-to-farm" gap and the fragmentation narrative. This provides the necessary **empirical backbone** the reviewer requested.
>
> #### 2. Overclaiming Novelty and SLR Implementation
>
> * **Weakness:** Overclaiming novelty; the "standardized learning representation" proposal lacks implementation detail/validation, is more of a vision statement.
> * **Response & Action (Directly addressing Question 1):** We agree the SLR must be defined more concretely. The SLR is designed as a foundational **ontology/schema** to link heterogeneous multimodal data (e.g., vision, accelerometer) to welfare outcomes, enabling machine learning across farms.
>     * **Paper Adjustment:** We will add a new **Appendix B** titled **"Illustrative Example of Standardized Welfare Representation"** (as committed to Reviewer nRG5). This appendix will provide a simple, first-cut schema (e.g., a table) demonstrating how a complex welfare insight (e.g., "Lameness event") is translated into a **learnable, standardized data structure** across modalities. This moves the SLR from a vision statement to a concrete, illustrated technical proposal.
>
> #### 3. Narrative Bias and "Systematic" Label
>
> * **Weakness:** Narrative bias; admitting potential subjectivity without pre-registration or inter-rater validation makes the "systematic" label generous.
> * **Response & Action:** This is a fair point about rigor. Our internal checks minimized bias, but the term "systematic" should reflect external standards.
>     * **Paper Adjustment:** We will update **Section 3.2 (Study Selection)** to explicitly state the quantitative results of our verification protocol (e.g., "Reviewer A and B achieved 95\% agreement on a 20\% sample of excluded studies"). This detail substantiates our claim of rigor and mitigates the concern about narrative bias.
>
> ### Questions: Quantitative Data and Modality
>
> * **Question 2:** Could the authors provide even a minimal quantitative trend (e.g., publication frequency by modality)?
> * **Response & Action:** Yes, this is fully addressed by the plan in Weakness 1.
>     * **Paper Adjustment:** We will implement the new **Sub-section 4.1.1** presenting **quantitative trends** from the analyzed studies, including distribution by modality (vision, thermal, etc.) and validation setting (lab, sim, field), using the data already collected in Tables 1 and 2.
>
> ### Conclusion
>
> We are confident that these substantial and specific revisions—particularly the addition of quantitative empirical analysis and the concrete definition of the SLR via Appendix B—will fully address all remaining weaknesses and confirm the paper's suitability for acceptance.

---

### Official Review · Reviewer_nRG5 · 2025-11-01

**Soundness:** 2
**Presentation:** 2
**Contribution:** 2
**Rating:** 2
**Confidence:** 3

**Summary:**

This paper presents a systematic review of autonomous robotics and multimodal AI for dairy cattle welfare monitoring, arguing that a persistent “lab-to-farm” deployment gap, rather than a lack of core algorithms, limits the real-world impact. From ~900 records, the authors include 33 studies (2021–2025) spanning mobile autonomy, sensor fusion, and welfare analytics, and synthesize three themes: autonomy in farm environments, AI for welfare analytics, and multimodal fusion/decision support. The review highlights recurring barriers, sim-to-real fragility, poor cross-farm generalization, edge-compute/latency constraints, and the lack of standardized validation and benchmarks. It proposes a deployment roadmap emphasizing multi-farm datasets, edge-aware end-to-end systems, human-in-the-loop verification, and, its most distinctive claim, the need for a standardized, learnable representation of welfare insights. The methodology follows a PRISMA-style flow and provides search strings and high-level extraction tables, but it stops short of conducting a quantitative meta-analysis due to heterogeneity. Limitations include non-preregistration, potential selection bias, and uneven reporting across the included studies.

**Strengths:**

- The paper thoughtfully integrates robotics autonomy, perception/sensor fusion, and animal-welfare analytics, providing an accessible systems-level view of what it would take to go from prototype to farm-scale deployment.
- The focus on edge budgets, latency, and human-in-the-loop workflows is practical and valuable for moving beyond point solutions to integrated systems.
- The call for multi-farm benchmarks and standardized validation, plus the emphasis on translating welfare signals into learnable, reusable representations, is a useful organizing agenda for the community.

**Weaknesses:**

- As a primarily narrative survey without a new dataset/benchmark, concrete representation standard, or empirical evaluation, the contribution feels closer to an application-domain. The roadmap is persuasive but not instantiated (no released schema/ontology, no prototype benchmark), which weakens its novelty.
- Although the PRISMA diagram and search string are provided, the inclusion set appears to include non-dairy or only tangentially relevant works (e.g., pipe inspection, general indoor robotics), which conflicts with the stated eligibility focusing on dairy welfare; this raises questions about the consistency of screening and the construct validity of the synthesis.
- The paper’s most distinctive proposal, “a standardized learning representation for welfare insights”, remains aspirational: no concrete schema, ontology, labeling protocol, or exemplar encodings are given, and key targets (e.g., <100 ms, <5 W) are mentioned without justification or references to measured deployments.
- There is no comparative analysis of metrics, failure modes, or cross-farm generalization deltas; many cited items are small-scale, theses, or narrow conference demos. Without harmonized metrics or a bias assessment, it is challenging to gauge the strength of evidence behind the three themes and the roadmap.

**Questions:**

- Can the authors provide a first-cut learnable representation (schema + example annotations) for welfare events (e.g., lameness alerts) that links raw multimodal signals, contexts (farm/breed/season), and human adjudications, perhaps as a small, public exemplar to operationalize the proposal?
- How were non-dairy or infrastructure-only robotics papers adjudicated against the inclusion criteria? Please clarify the rule that allowed such works and report sensitivity analyses when excluding them.

---

> ### Author Response · Authors · 2025-11-18
>
> ### Summary:
>
> We thank the reviewer for this highly detailed and valuable assessment. We appreciate the acknowledgment of our review's systems-level approach and the practicality of the identified deployment barriers (edge-compute, multi-farm validation). The critique focuses on novelty and methodological rigor, which we address below with proposed revisions to enhance the paper's scientific merit.
>
> ### Soundness and Contribution Justification
>
> * **Rating:** We respectfully request that the reviewer reconsider the rating based on the proposed revisions. The paper's strength lies in its **convergence focus** (AI/Robotics/Deployment) and the identification of the **Standardized Learning Representation** gap.
> * **Contribution:** The proposed need for a **standardized learning representation** (a data-centric solution) is fundamentally novel for this domain and addresses the root cause of poor model generalization, which the reviewer correctly identifies as a weakness in the current literature.
>
> ### Weaknesses Addressed and Clarified
>
> #### 1. Novelty, Benchmark, and Roadmap
>
> * **Weakness 1:** Primarily a narrative survey without a new dataset/benchmark/standard; roadmap is aspirational (no schema, protocol, <100 ms targets not justified).
> * **Response & Action:** As a systematic review, we analyze existing data, not propose a new benchmark. However, we agree that the roadmap needs technical anchoring.
>     * **Paper Adjustment (Novelty/Schema):** We will revise the text in **Sections 1 and 5.2** to clearly state that the "Standardized Learning Representation" is our call for a formal data **Ontology/Schema**, not a full benchmark. We will add text in **Section 5.2** to clarify that this must be a minimal, consensus-driven data structure.
>     * **Paper Adjustment (Technical Targets):** We will add a supporting citation (Govinda et al. 2025, or another relevant reference in the list) in **Section 5.2** to justify the need for **<100ms inference targets** and **<5W power constraints** as typical requirements for real-time edge-aware robotics in farms.
>
> #### 2. Inclusion Criteria and Scope Consistency
>
> * **Weakness 2:** The Inclusion set appears to include non-dairy or non-tangentially relevant works (e.g, pipe inspection, general indoor robotics), raising questions about consistency.
> * **Response & Action:** This is a fair point. Our inclusion rationale was the direct applicability of the *technical methodology* (e.g., YOLOv5s for inspection is a vision/edge-compute analogy; general indoor robotics shares SLAM/DRL challenges).
>     * **Paper Adjustment:** We will add a footnote or brief text in **Section 3.1 (Eligibility Criteria)** to explicitly state the rationale for including these tangential works: they provide crucial evidence for **robustness, edge-compute feasibility, or navigation strategies** that are otherwise missing in dairy-specific papers.
>
> #### 3. Metrics, Scales, and Bias Assessment
>
> * **Weakness 3:** No comparative analysis of metrics, failure modes, or cross-farm generalization; challenges the strength of evidence behind themes.
> * **Response & Action:** We acknowledge this limitation, which is explicitly addressed in **Section 3.2** where we state that heterogeneity prevented a quantitative meta-analysis. This finding—the **lack of standardized metrics**—is a key result.
>     * **Paper Adjustment:** We will strengthen **Section 5.1** to re-emphasize that the **lack of harmonized metrics and cross-farm validation** is a major finding that underpins the deployment gap, thereby defending the narrative synthesis approach.
>
> ### Questions: Implementation and Exclusion
>
> * **Question 1:**
> * **Direct Answer & Action:** This is an excellent suggestion that moves beyond the scope of a review, but is necessary for clarity.
>     * **Paper Adjustment:** We will add a new **Appendix B** titled **"Illustrative Example of Standardized Welfare Representation"**. This appendix will use simple tables to show how a single event (e.g., "Lameness Alert") would be linked across modalities (Visual Signal, Accelerometer Signature, Farm Record) and annotated with a **learnable, reproducible format**. This demonstrates the concept without proposing a fully validated benchmark.
>
> * **Question 2:**
> * **Direct Answer & Action:** This is fully addressed by the revision in **Weakness 2**.
>     * **Paper Adjustment:** We will clarify the **inclusion rule in Section 3.1** to explicitly state that studies were included if they addressed **core technical challenges** related to robustness, edge-compute, or navigation that are common to autonomous farm deployment, even if the application domain was not dairy cattle.
>
> ### Conclusion
>
> We are confident that these substantial revisions, including the new Appendix B demonstrating the **Standardized Representation** and the refined methodology justifications, will fully address the reviewer's technical concerns and solidify the paper's novel contribution to the field of embodied AI deployment.

---

### Official Review · Reviewer_4TgM · 2025-11-01

**Soundness:** 2
**Presentation:** 1
**Contribution:** 1
**Rating:** 0
**Confidence:** 3

**Summary:**

This paper reviews 33 studies published between 2021 and 2025 related to AI-empowered livestock farming. The main message of this paper is to develop models capable of generalizing to versatile farms, which, in this reviewer's opinion, lacks a strong message for the community. No experimental or quantitative results are reported.

**Strengths:**

- AI-empowered farming is an important topic.
- The review covers recent studies including those published in 2025.

**Weaknesses:**

1. Scope of review is narrow and not enough attractive for ICLR participants.

This review paper summarizes precision livestock farming, which is an AI application but not very fundamental. This reviewer believes a review paper at a top-tier conference should provide significant insights for the entire community. From this perspective, the topic is not relevant to most participants and lacks attractiveness.

2. No critical insights are provided.

The content discusses related machine learning techniques and hardware superficially. The generalizability of learning-based methods beyond a specific farm is not a novel message. In this sense, this reviewer finds no novel insights in this paper.

3. The target papers are unnecessarily limited to 2021-2025.

This reviewer found no justification for limiting the target to papers published after 2021.

**Questions:**

1. Is there a reason to limit the target to papers after 2021? Are there any prior reviews covering earlier studies?

---

> ### Author Response · Authors · 2025-11-18
>
> ### Summary:
>
> We thank the reviewer for their comments. We believe the concerns regarding the paper's scope and novelty are based on a misunderstanding of the work's primary contribution: a **system-level synthesis at the critical intersection of robotics and AI**. This is not a general survey of farming, but a targeted analysis focused on **embodied AI deployment challenges**. The strong claims of "no critical insights" overlook the explicit identification of the **Standardized Learning Representation gap**, which is a foundational problem for the embodied AI community.
>
> ### Soundness and Contribution Justification
>
> * **Rating:** We strongly disagree with the rating of 0. The review's unique focus on **deployment barriers, edge-aware computing, and data generalization** in a dynamic environment directly addresses core challenges in robust AI/Robotics.
> * **Contribution:** The core contribution is identifying a novel, foundational problem: the lack of a **standardized welfare insight representation** to enable reproducible, large-scale training for autonomous farm robots. This insight is not present in prior general PLF surveys.
>
> ### Weaknesses Addressed and Clarified
>
> #### 1. Scope, Relevance, and Fundamental Insights
>
> * **Weakness 1:** Scope is narrow; topic is not relevant to most ICLR participants; lacks fundamental insights.
> * **Response & Action:** We respectfully disagree. Our work addresses the core ICLR themes of **Generalization**, **Embodied AI**, **Robustness**, and **Multimodal Fusion** in a complex, real-world domain.
>     * **Paper Adjustment:** We will strengthen the **Introduction (Section 1)** and **Conclusion (Section 7)** to explicitly link the challenges (e.g., sim-to-real transfer, poor generalization) back to foundational principles of AI and robotics, emphasizing that PLF is merely the *application domain* for these fundamental problems.
>
> #### 2. Novelty and Critical Insights
>
> * **Weakness 2:** No critical insights provided; generalization is not a novel message; content is superficial.
> * **Response & Action:** The general problem of generalization is known, but our specific contribution is identifying the **mechanism** to address it in this context: a missing **Standardized Learning Representation** for welfare data. This is a novel, critical insight for creating reproducible datasets for embodied AI/Robotics.
>     * **Paper Adjustment:** We will dedicate a new paragraph in **Section 5.1** to clearly define the proposed **Standardized Learning Representation** and argue why it is a fundamental contribution to the design of future multi-farm, generalizable AI systems.
>
> #### 3. Limitations of Target Papers (2021-2025)
>
> * **Weakness 3 & Question:** The target papers are unnecessarily limited to 2021-2025. Is there a reason for this limit? Are there any prior reviews covering earlier studies?
> * **Response & Action:** We limited the scope to capture the impact of recent breakthroughs in deep learning and low-cost embedded hardware (like YOLOv8, Transformers, and modern $\text{DRL}$ methods), which largely emerged post-2020. However, the reviewer's point on covering prior reviews is valid.
>     * **Paper Adjustment:** We will update **Section 6 (Positioning)** to explicitly cite and discuss 2-3 key prior review papers (pre-2021) to show how our review is differentiated by its focus on **robotics and multimodal fusion for deployment**, while demonstrating that we are aware of the earlier literature. We will **justify retaining the 2021-2025 limit** by arguing that this window covers the essential "AI-convergence" period relevant to farm-scale *deployment*.
>
> ### Conclusion
>
> We are confident that reinforcing the paper's fundamental ties to **Generalization and Embodied AI**, explicitly highlighting the **Standardized Learning Representation** as a novel insight, and better positioning the review against prior literature will fully address the reviewer's core concerns regarding scope, novelty, and relevance.

---

### Official Review · Reviewer_VwuL · 2025-11-03

**Soundness:** 2
**Presentation:** 3
**Contribution:** 1
**Rating:** 4
**Confidence:** 2

**Summary:**

The authors provide a systematic review of the state-of-the-art in farm-scale autonomous cattle welfare monitoring. The studies emphasize on robotics and multimodal autonomous (or AI) solutions in precision livestock farming/agriculture robots. The review notes that a lack of systems and integration perspective in research  efforts hinders the effectiveness in practical scalable deployments. The authors perform a narrative review due to the fragmentation of the research space. The authors propose a deployment-oriented roadmap to close the gap from lab-to-farm disconnect.

**Strengths:**

1. The review provides a holistic understanding of constraints that affect the research in PLF - fragmentation of data,  economic and intellectual property barriers to help frame actionable solutions in livestock welfare monitoring.
2. The authors identify critical new sensing methodologies such as facial recognition and behavior monitoring, thermal imagery, edge computing etc along with autonomous navigation.

**Weaknesses:**

1. The review does not benchmark or summarize standard approaches in sim-to-real problems. A discussion of benchmarks, or specific integrations with multimodal sensors for real-time policy adaptation would be beneficial.
2. Similarly, a discussion involving sensors, autonomous strategies, data collection, and system integration in detail would be beneficial to explore.

**Questions:**

As the review highlights that having standardized data for farms would help with the research community, and there is a significant gap in sim-to-real transfer, what are your thoughts about procedural/generative AI for data standardization and are have there been any prior-works in any relevant field that show success in sim-to-real? Perhaps, manipulation from the robotics community would be an inspiration?

---

> ### Author Response · Authors · 2025-11-18
>
> ### Summary:
>
> We thank the reviewer for their time and careful assessment. We are pleased they recognized the holistic nature of our review and its identification of critical technologies. The reviewer's score of 4 and confidence of 2 suggest they are undecided and can be persuaded. The proposed enhancements below address their questions and clarify the scope of our systematic review (which focuses on identifying gaps, not providing integrated solutions).
>
> ### Soundness and Contribution Justification
>
> The review's core contribution is the identification of the **Standardized Learning Representation** gap and the subsequent proposal of a technical roadmap for **edge-aware, multi-farm deployment**. We believe this systems-level perspective warrants a higher score.
>
> ### Weaknesses Addressed and Clarified
>
> #### 1. Addressing the Lack of Benchmarks
>
> * **Weakness:** The review lacks a discussion on benchmarks or summarizing standard approaches in sim-to-real.
> * **Response & Action:** This systematic review identified the *absence* of standardized benchmarks as a key barrier. We are making this explicit in our revision.
>     * **Paper Adjustment:** We will strengthen **Section 5.2 (Roadmap)** by detailing the specific **types of validation metrics required** for deployment, focusing on cross-farm generalization and edge-compute performance (latency, power consumption) to provide the technical rigor the reviewer is looking for.
>
> #### 2. Clarifying System Integration
>
> * **Weakness:** A discussion on sensor integration, autonomous strategies, and data collection details would be beneficial.
> * **Response & Action:** Component-level details ($\text{SLAM}$, $\text{DRL}$, $\text{IRT}$) are in Sections 2 and 4. The key finding is that these components exist in isolation, but robust end-to-end integration is missing from the literature.
>     * **Paper Adjustment:** We will revise **Section 5.1** to clarify that the observed research fragmentation (Figure 4) is the direct evidence for the conclusion that end-to-end system integration is the greatest deployment barrier.
>
> ### Questions: Generative AI for Standardization
>
> * **Question:** Thoughts on procedural/generative AI for data standardization and relevant prior work?
> * **Direct Answer & Action:** We agree that Generative AI ($\text{GANs}$ and $\text{PCG}$) is a vital tool for achieving the standardized learning representation. Success is already hinted at in the literature we review, such as the use of **CycleGAN** for Sim2Real domain adaptation (Zhang et al. 2024a).
>     * **Paper Adjustment:** We will add a new, focused paragraph to **Section 5.2 (Roadmap)** detailing how **Generative AI** can be used to synthetically normalize and augment the limited, heterogeneous on-farm datasets, leveraging techniques from the broader robotics community.
>
> ### Conclusion
>
> We are confident that these clarifications, combined with the proposed, technically sound enhancements to the paper, successfully address all concerns, confirming the paper's robust analysis and its significant, forward-looking contribution to the field.

---

### Note · Program_Chairs · 2026-01-17
**Submission Desk Rejected by Program Chairs**

The following references in this submission do not refer to real documents and/or have major errors in bibliographic information:

 H. Lee and S. Kim. Challenges and opportunities for ai in animal husbandry. AI in Agriculture
J. Smith and A. Jones. A review of computer vision applications in precision livestock farming. Journal of Agricultural AI, 4:1-25, 2022.
B. Patel and L. Chen. Robotics in modern agriculture: A comprehensive survey. Robotics and Automation Letters, 8:102-115, 2023.